# Ecotourism as a Forest Conservation Tool: An NDVI Analysis of the Sitakunda Botanical Garden and Ecopark in Chattogram, Bangladesh

**Nazifa Rafa** [1,2,3], **Samiha Nuzhat** [1,4], **Sayed Mohammad Nazim Uddin** [1,2,*], **Mukesh Gupta** [1,2] **and Rahul Rakshit** [5]

1   Environmental Sciences Program, Asian University for Women, 20 M. M. Ali Road, Chattogram 4000, Bangladesh; nazifa.rafa@post.auw.edu.bd (N.R.); samiha.nuzhat@post.auw.edu.bd (S.N.); mukesh.gupta@auw.edu.bd (M.G.)
2   Center for Climate Change and Environmental Health (CCCEH), Asian University for Women, 20 M. M. Ali Road, Chattogram 4000, Bangladesh
3   Department of Land Economy, University of Cambridge, The Old Schools, Trinity Lane, Cambridge CB2 1TN, UK
4   Water and Life Bangladesh, H-2, Road-28, Block-K, Banani, Dhaka 1213, Bangladesh
5   Esri, 380 New York Street, Redlands, CA 92373, USA; RRakshit@esri.com
*   Correspondence: sayed.uddin@auw.edu.bd

**Abstract:** Ecotourism, a sustainable form of tourism, is increasingly being viewed as a tool that can promote global biodiversity and forest conservation. This study explored the scope of ecotourism in forest conservation practices in the developing context by taking the Sitakunda Botanical Garden and Ecopark (SBGE), Bangladesh's first ecopark established in 2000, as a case study. Using GIS and remote sensing technology, NDVI analysis revealed that, unlike the anticipated outcomes of the SBGE project, after a brief increase in vegetation coverage of 84.6% from 1995 to 2000, the vegetation coverage fell drastically from 2000 to 2015, wherein 33.4% of vegetation had been completely removed, and much of the dense and medium vegetation had been converted to sparse vegetation or other land uses. Anthropogenic activities, namely, unplanned urbanization, are suggested as the major contributors to this decline. From the period of 2015 to 2020, however, vegetation was seen to regenerate, potentially due to the decelerating urbanization or the possible manifestation of the 'U' shape relationship between the changes in vegetation and rates of urbanization. Sustainable land-use policies may help attain the targets of the project and lead the SBGE to emerge as a success story of the Bangladeshi ecotourism industry.

**Keywords:** Bangladesh; forest conservation; ecotourism; GIS; NDVI; sustainable tourism; urbanization; vegetation

## 1. Introduction

The tourism industry has become one of the fastest-growing industries in the service sector, wherein between 2009 and (pre-pandemic) 2019, the real growth in international tourism receipts (54%) exceeded the growth in world GDP (44%), contributing USD 1481 billion in total international tourism receipts alone [1]. Even though the COVID-19 pandemic significantly lowered tourism across the world, having caused a drop of 73% in international global tourist arrivals in 2020, international tourism experienced signs of a rebound in June and July 2021, attributed to the easing travel restrictions and the advancing global vaccination rollout [2].

Tourism can be of different kinds, usually deriving its characteristics from the intentions of the tourist. One such facet of tourism is ecotourism. The term 'ecotourism' was first coined by Ceballos-Lascuráin in the early 1980s, wherein he defined it as 'traveling to relatively undisturbed or uncontaminated natural areas with the specific objective of studying,

admiring, and enjoying the scenery and its wild plants and animals, as well as any existing cultural manifestations (both past and present) found in these areas' [3] (p. 17). Ecotourism primarily emerged as the need for sustainable tourism and was recognized since mass tourism often constituted uneven development and high social and environmental costs [4]. It is increasingly being viewed as a potential tool that can bring about sustainable development [5–7], as it is concerned with sustainable forms of tourism that take place in the natural areas, and promotes environmental conservation, environmental awareness, travelers' responsibility, and active community participation [8]. It allows for the reconciliation of both economic growth and environmental wellbeing, as ecotourism generates revenue while simultaneously encouraging initiatives for the conservation and the management of biodiversity [9,10]. Therefore, ecotourism also has wide-ranging implications in global biodiversity conservation initiatives.

Ecotourism can serve as an important tool for sustainable development, especially in developing nations that possess impressive biodiversity hotspots [4,11–13]. For instance, Bangladesh, a South Asian developing country with a population of over 160 million people, derives its potential for ecotourism by possessing several world-famous natural sites, such as the Sundarban and the Cox's Bazar, where it boasts not only areas containing spectacular jungles rich in wildlife, waterfalls, rivers, and hilly landscapes, but also several cultural heritage sites [14,15]. Despite such positive connotations, ecotourism still remains a highly contentious concept [4], particularly due to its reliance on market-based conservation making [16,17]. As the implications of such limitations are far more significant in nations that are struggling to operationalize sustainable development, it is important to assess the impacts of ecotourism interventions in such countries. This section starts by providing a brief overview of how ecotourism can be utilized to attain biodiversity conservation, revealing both its strengths and weaknesses in the endeavor. It then introduces readers to the study area of interest and delves into discussions of how remote sensing and spatial analytical techniques have been contributing to ecotourism research, especially in developing nations. Finally, it outlines the objectives of this study and provides insights into the novelty of this research.

### 1.1. Ecotourism as a Tool for Biodiversity Conservation

Ecotourism practiced in areas with impressive biodiversity and landscapes is a promising sub-sector of tourism [18,19] but has often been criticized for being ineffective and/or harmful, initiating numerous environmental risks such as water pollution and old-growth deforestation due to the increasing reliance and usage of natural resources, especially forest products when the ecotourism spot is or nearby a forest [20–22]. In many case studies, activities related to tourism increased the demand for timber and fuelwood for the construction of new infrastructure, including housing [20]. The economic development of the area also acted as a pull factor for migration and population growth, and many forested lands were cleared for other land uses [20]. However, most of the claims opposing ecotourism are attributed to flawed research designs when studying the topic of ecotourism, making it difficult to assess the simultaneous economic, environmental, and social benefits it offers [21]. On the contrary, ecotourism has been reported to lead to forest regeneration, particularly in agrarian landscapes, when approached with conservation mechanisms, such as protected areas, Payment for Ecosystem Services (PES), and monitoring/enforcement [20]. For instance, when a PES system was instituted at the Monarch Butterfly Biosphere Reserve in Mexico, more gains than losses were experienced in the closed cover density during 1999–2009 [23]. Therefore, ecotourism can be viewed as an incentive-driven forest governance intervention. In addition, by creating the perception of biodiversity as 'economic goods' [24], ecotourism can bring advantages in conservation by supporting wildlife and protected areas, diversifying livelihoods, promoting environmental interpretation and ethics, and strengthening resource management [21]. The income generated from ecotourism can be also used for the landscape-scale conservation of habitats for a diverse group of animals and plants [18,25]. Several ecotourism initiatives, such as

the Chitwan National Park in Nepal [20,26], are illustrations of ecotourism as a promising forest conservation tool. In the Chitwan National Park, satellite image analysis displayed regeneration of many forest patches after the introduction of a buffer zones program due to significant investment in plantation and forest-management initiatives.

Nevertheless, current practices of ecotourism run the risk of prioritizing environmental protection over local community welfare [27], wherein the lack of socio-cultural development and community participation is the primary cause of the failure of ecotourism as a tool for forest and biodiversity conservation and sustainable development. Many areas promoting ideas of ecotourism have initiatives conflicting with the daily practices of local communities [28,29]. In addition, the concept of ecotourism may not reach all levels of a community [30]. As a result, the importance of indigenous-based ecotourism driven by local participation is increasingly being recognized [31,32] and has proven to be successful [33].

### 1.2. The Untapped Potential of Ecotourism in Bangladesh

In Bangladesh, the total contribution of travel and tourism is expected to increase from 4.1% (statistics of 2014) to 6.5% by the year 2025 [34]. Currently, several laws and acts guide the ecotourism industry in Bangladesh, such as the Ecotourism Development and Management Plan 2004 and the National Tourism Policy 2010 [15]. Despite the internationally congruent guidelines provided by the aforementioned laws and acts, and Bangladesh Forest Department (BFD)'s current plans to improve and develop protected areas and ecological parks (ecoparks, areas which serve as leisurely parks without affecting the natural environment) [35], ecotourism is not properly operationalized in Bangladesh. Apart from the poor tourist experience [36–38], a lack of responsible tourist behavior can cause environmental pollution and degradation [36,39]. The principles of ecotourism are also disregarded due to the lack of encouragement of participation from locals and the lack of awareness of the concept of ecotourism [14,15,37]. In addition, a high visitor count, beyond the carrying capacity of the ecological area, is likely to hamper the sustainability of the resources [40]. The lack of economic benefits for locals and environmental degradation perceived by locals in some ecotourism spots are also likely to hinder initiatives from being successful [41,42].

### 1.3. The Sitakunda Botanical Garden and Ecopark

The Sitakunda Upazila in the Chattogram district of Bangladesh has been developed as a satellite town to tackle the population strain in Chattogram city, where the Upazila also serves as a zone for industrial development, driven by the Dhaka-Chittagong Highway and the railway. Even though agriculture dominates the livelihoods of the people in Sitakunda, economic development is led by the ship-breaking industry, which is currently the largest in the world. Sitakunda also possesses the nation's first ecopark, known as the Sitakunda Botanical Garden and Ecopark (henceforth SBGE, or Botanical Garden). Tourism at the SBGE has risen over the years, where recreational activities, educational purposes, and religious activities have been identified as the major objectives behind site visitation [43].

The SBGE was established on 808 ha of land at the Sitakunda Upazila in 2000 under the Bangladesh Wildlife Preservation (Amendment) Act 1974 as a bid to conserve the rich biodiversity of the area, of which the botanical garden covers 405 ha and the rest is the ecopark. Several governmental and religious infrastructures and three natural waterfalls are located inside the park. The SBGE was an initiative under the five-year project of the Ministry of Environment and Forest (MoEF) which aimed to: (1) expand, preserve, and develop the existing biodiversity of the indigenous species through intensive management, (2) cultivate and preserve various species of bamboo, cane, herbs, and medicinal plants, (3) initiate biodiversity conservation, improve habitats of wildlife, and protect endangered wildlife, (4) build infrastructure to promote ecotourism, and (5) build research and education facilities [44].

Prior to the establishment of the SBGE, the semi-evergreen sub-tropical forest had been lying denuded for years. Encroachment, illegal felling, and the ravages of wars of 1941–1945 and 1971 removed 21,000 ha of forests, which reduced the soil fertility by increasing soil erosion, decreasing water retention, and promoting runoff and compaction of topsoil [45]. However, the protected status of the SBGE has led to the regeneration of many dwindling floral species. It was only after recent protection measures that indigenous species could regenerate naturally [45]. The SBGE hosts a total of 412 vascular plant species under 315 genera belonging to 94 plant families [46]. The majority of the plant species are herbs, followed by trees, shrubs, and climbers, and many of these are exotic plant species [46,47]. The park authorities also create plantations of native species every year [45]. Over the years, the total crown coverage and biodiversity have increased remarkably, whereas soil erosion has been reduced [48]. The SBGE now consists of high forests, low forests, grasslands, and water bodies, where the high and low forests are the major habitat types [48].

*1.4. Application of GIS in Ecotourism Research and Vegetation Cover Analysis*

Analysis of spatial data is increasingly becoming easier and effective thanks to geographic information system (GIS) and remote sensing (RS) technologies. Integration of both GIS and RS tools provides greater advantages for object-oriented spatial data modeling, as the data produced are far more accurate [49,50]. Thus, the combined GIS-RS technology has implications for research related to health, natural hazards, environmental issues, social issues, or any spatial, temporal, or spatiotemporal analysis [51–53]. GIS-RS technology provides scope for researchers to pursue analysis and re-interpretation of spatiotemporal information obtained according to their interests and is particularly a huge advantage for researchers working in developing countries, where demographic data, research funds, and innovative technologies for spatial research are scarce [54].

The application of GIS in ecotourism research has been far and wide. GIS has primarily been used to identify and evaluate the scope of ecotourism in natural environments across various nations and ecosystem zones, including in several developing nations, often in combination with various other frameworks [55–59]. In Bangladesh as well, GIS has been used to propose ecotourism spots in Sundarban [60] and Cox's Bazar [61]. Unfortunately, the planning of SBGE did not experience the benefits of using GIS to glean the suitability of the area for ecotourism purposes.

The application of GIS has also been recommended in the assessment of biodiversity and forest conservation practices [62–64]. The archetypal case study for the negative forest outcomes of ecotourism, for instance, was illustrated by Liu et al. [65] in the analysis of the impacts of tourism on the Wolong Giant Panda Nature Reserve in southwest China. The study revealed that there was an increase in deforestation after the Reserve was implemented, and found that, surprisingly, deforestation occurred more within the Reserve than outside. Similar studies were conducted in other ecotourism spots in China, and other developing nations like India, Nepal, Cambodia, Mexico, Belize, and Peru [20]. The most recent literature [66] focused on the land use and cover of the Shivpuri watershed in Nepal, also repeating a similar scene: a 110 ha reduction in forest coverage between 1999 to 2016. However, the application of GIS and/or RS technology to assess the effectiveness of biodiversity conservation strategies, such as in protected areas, has been less extensive in Bangladesh. Nevertheless, one study found that the Himchari National Park, a protected area, has been degraded, fragmented, and converted severely into various land uses, wherein nearly half of the dense forest land was converted to other land uses in the period of 1977 to 2017 [67].

Current methods of biodiversity conservation assessment involve tedious traditional, time, and resource-consuming approaches such as surveys, sampling, observation, etc. However, spatial analytical tools make land-use and land-cover pattern analysis more cost-effective, easier, accurate, and quicker [68], and effectively provide information on land-cover changes, even in humid tropical areas, which can support environmental

monitoring and government development programs [69]. Therefore, this study draws in the strengths arising from the integration of GIS and RS technology and serves as a tool to assess the progress of biodiversity conservation practices.

*1.5. Objectives*

Carbon sequestration and the growth of carbon 'sinks' have been the objectives of many of the existing projects of the MoEF [35]. Despite contradictions on the total forest coverage in Bangladesh [70], the BFD reports a national forest coverage of 2.53 million ha, spanning 17.49% of the total land of the country [35]. With the government of Bangladesh, and particularly the MoEF, striving to increase the resilience of forests and protected areas and undertaking initiatives to mitigate and adapt to climate change [35], it is crucial to understand how initiatives and ambitions of the SBGE have affected forested land coverage in the area as forestry is being targeted as the major climate change mitigation strategy by the MoEF. Most scholarly works have attempted to assess the flora and fauna biodiversity of the SBGE, but a spatiotemporal analysis of how the vegetation coverage has changed has not been attempted before for the SBGE. Accordingly, the objectives of this study were to explore how effectively ecotourism and, consequently, a protected status served as a forest conservation tool by assessing the vegetation cover changes from 1995 to 2020 in and around the SBGE by employing GIS-RS technology. This study then attempts to suggest and conceptualize plausible anthropogenic activities that may have contributed to the changes in vegetation by providing insights into the tourism practices in and around the SBGE, and the socio-economic activities of the local community via a comprehensive literature review of relevant studies surrounding the SBGE.

This study derives its novelty from the relatively unexplored nature of ecotourism in Bangladesh in both research and implementation, leading to various structural and institutional flaws in ecotourism initiatives. Particularly, the potential of biodiversity conservation of ecotourism is relatively underappreciated, especially in Bangladesh. Moreover, because of the relatively limited use of GIS in ecotourism and biodiversity conservation research in Bangladesh, this study revealed the benefits and limitations of GIS and remote-sensing technology in ecotourism research and conservation initiative assessment in Bangladesh.

## 2. Methodology

Overall, there is a dearth of literature that empirically analyzes ecotourism impacts on forests. Brandt and Buckley [20] found only 17 studies, published between 2000 to 2018, which evaluated the potential of ecotourism for forest protection in biodiversity hotspots. The majority of the studies (14 of 17) evaluated forest change using satellite data, while the remaining used methods from social sciences (such as surveys and interviews) and regression models to support the association. The Wolong Giant Panda case study, for example, represents a clear understanding of how the coupling between human and natural systems varies across spatial units. The understanding of the complexities between human and natural systems is often hindered by the academic separation of ecological and social sciences [71]. Therefore, an interdisciplinary perspective comprising ecological and social sciences is crucial to study coupled human and natural systems [71]. As mentioned, this study employed GIS-RS techniques to assess the forest conservation initiatives in the SBGE. However, as this study was conducted when the pandemic situation was serious, with high rates of death and infectivity and intensified mobility restrictions (early 2020–early 2021), no primary data could be collected to complement the findings from the GIS-RS analysis (more in Section 3.3. Limitations).

This section describes the methods that were undertaken by this study in detail. Figure 1 below provides the summary of the workflow of the vegetation coverage analysis conducted for this study.

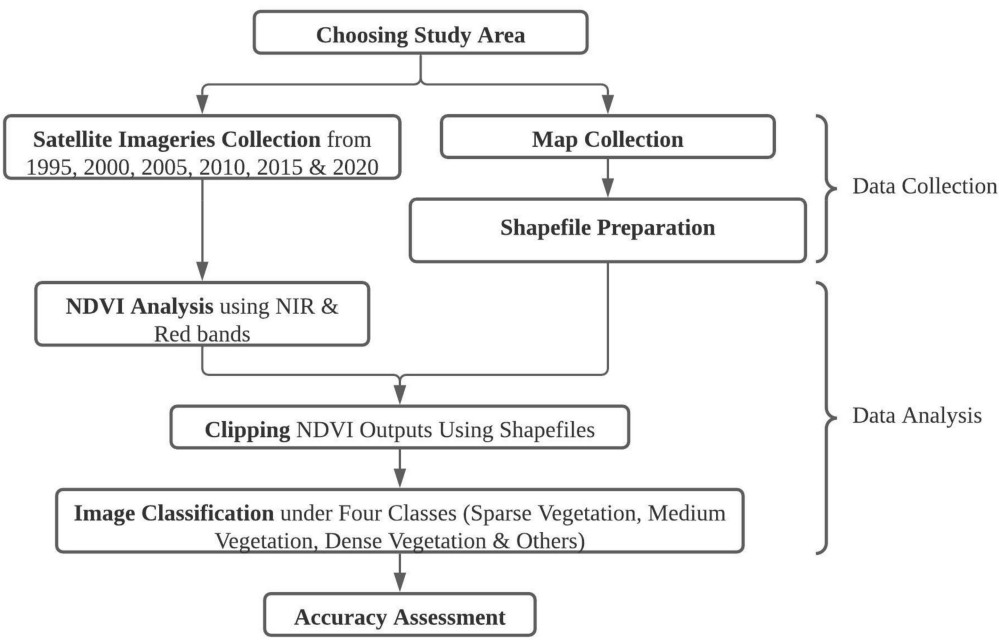

**Figure 1.** A Schematic Diagram of the Workflow of the Vegetation Coverage Analysis.

*2.1. Study Area*

The SBGE is situated at the Sitakunda Upazila in Chattogram District, Bangladesh. The SBGE comprises the Chandranath reserve forest, which is under the jurisdiction of the Chittagong Forest Division and lies between 22°36′–22°39′ N and 91°40′–91°42′ E. It is situated in the topography of medium-high to low hill ranges with an altitude of 352 m above sea level, where the hills are made of sandstone and shale [48]. The climate of the area is moist tropical, with a mean annual temperature of 26.6 °C, and much of the rainfall occurs from June to September, with the highest in July (596.6 mm on average) [48]. Apart from the forested natural environment that predominantly covers the SBGE, the primary sources of tourist attractions within the SBGE are the religious infrastructure and three waterfalls. Figure 2 shows the location of the SBGE in Chattogram, Bangladesh, and the rough outline of the SBGE.

The area of interest for this study also comprises the nearby area enveloping the SBGE to understand the effects of how the protected status of a specific forest area affects the nearby vegetation coverage.

*2.2. Satellite Imagery Collection*

Six Landsat satellite imageries of the study area, beginning from 1995 to 2020, with an interval of five years, were downloaded from an open platform, the Global Visualization Viewer (GloVis) website. Since the work for the establishment of SBGE was carried out in the year 2000, an earlier year (1995) was selected to explore the overall changes that the protection status has brought to the study area. The imagery specifications have been given in Table 1.

All the images collected had a cloud coverage of under 10%; images collected from the year 2000 onwards had a cloud coverage of less than 2%, but the image collected for the year 1995 had a cloud coverage of approximately 9% because images with less cloud coverage were not available for the year 1995. All images were taken during the daytime.

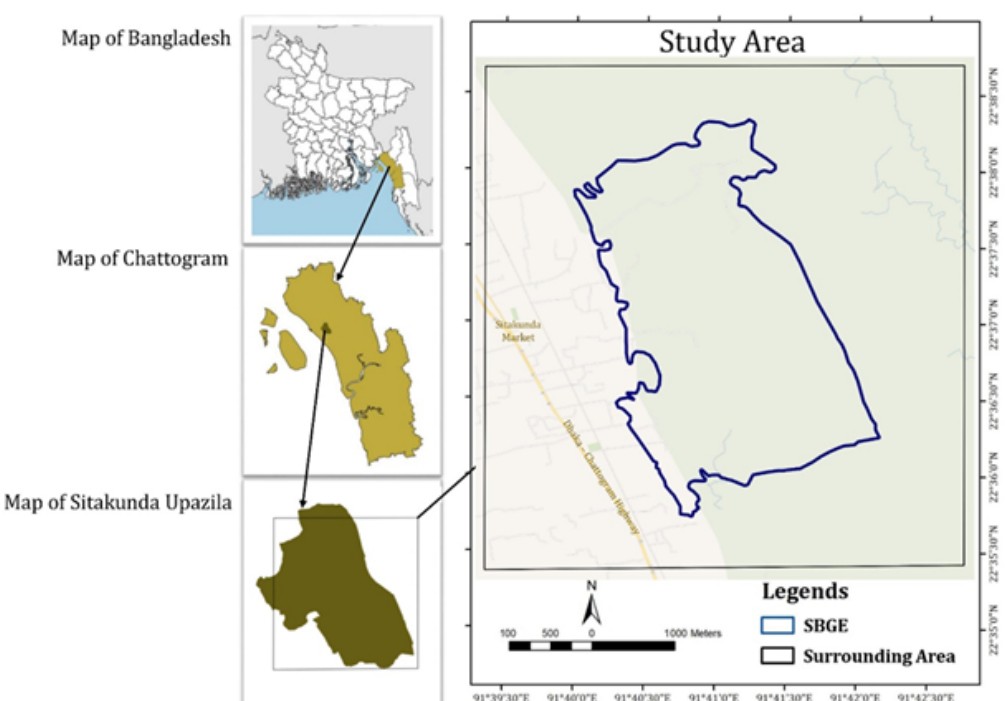

**Figure 2.** Location of the study area.

**Table 1.** Image specifications of the satellite images obtained for the study.

| No. | Date | Satellite | Path/Row |
|:---:|:---:|:---:|:---:|
| 1. | 30 January 1995 | Landsat 4-5 TM C-1 Level 1 | 136/44 |
| 2. | 12 January 2000 | Landsat 4-5 TM C-1 Level 1 | 136/44 |
| 3. | 10 February 2005 | Landsat 4-5 TM C-1 Level 1 | 136/44 |
| 4. | 8 February 2010 | Landsat 4-5 TM C-1 Level 1 | 136/44 |
| 5. | 26 March 2015 | Landsat 8 OLI/TIRS C-1 Level 1 | 136/44 |
| 6. | 19 January 2020 | Landsat 8 OLI/TIRS C-1 Level 1 | 136/44 |

*2.3. Shapefile Preparation*

In this study, a polygon shapefile was created to elucidate the boundary of the SBGE study area. The area of the SBGE shapefile is approximately $9.28 \times 10^6$ square meters. Thus, it is in accordance with the recorded area of the SBGE, which is 808 ha (or $8.08 \times 10^6$ square meters). A rectangular-shaped boundary was also drawn around the SBGE to assess the vegetation coverage of the surrounding area. The surrounding area spanned approximately $25.2 \times 10^6$ square meters. Therefore, the total area of the two constructed shapefiles was approximately $34.5 \times 10^6$ square meters.

*2.4. Data Analysis*

The collected satellite images did not require any pre-processing since the band-wise segregated raw satellite images were suitable for normalized difference vegetation index (NDVI) analysis. Data analysis of the study encompassed multiple stages which were conducted using three pieces of software: QGIS 3.16, ArcGIS, and ERDAS IMAGINE 2014.

2.4.1. Normalized Difference Vegetation Index (NDVI) Analysis

As illustrated in Table 1, the six sets of downloaded satellite images are from Landsat 4-5 TM and Landsat 8 OLI/TIRS. Consequently, the band combination for these two different categories of satellites is different. The vegetation coverage of the study area was detected using the Normalized Difference Vegetation Index (NDVI) classification

strategy. NDVI values denote simple graphical indicators that can be used to analyze remotely sensed data to assess the presence of live green vegetation [72]. By design, the NDVI varies between −1.0 and +1.0. Therefore, NDVI is functionally, but not linearly, equivalent to the simple infrared/red ratio (NIR/VIS). Areas with vegetation coverage generally yield high values for these indices due to their high near-infrared reflectance and low visible reflectance. The reflectance of cloud, snow, and water is larger in the red than in near-infrared regions. Clouds yield negative values while water, rocks, and barren land yield very low or slightly negative values. Therefore, an NDVI of zero or close to zero means no vegetation [73]. The formula for NDVI calculation is as below,

$$NDVI = \frac{NIR - Red}{NIR + Red}$$

where, NIR represents the Near Infrared band 4 (0.76–0.90 μm) of Landsat 4-5 and RED is the corresponding band 3 (0.63–0.69 μm). Similarly, for Landsat 8, NIR represents the Near Infrared band 5 (0.845–0.885 μm) and RED the corresponding band 4 (0.630–0.680 μm). Thus, the NDVI value of each pixel was detected only using the raster images of the aforementioned bands.

### 2.4.2. Supervised Classification

In this study, vegetation coverage change detection of the SBGE and its surrounding areas was analyzed for six years at an interval of 5 years. NDVI analysis is somewhat of an unsupervised classification as the software itself classifies the pixels under values ranging from −1 to +1. To detect the vegetation coverage, suitable ranges of NDVI values were declared. Here, areas containing dense vegetation tended to have positive values of 0.2 or above. The type of vegetation was further classified according to their relative densities. The raster images obtained through NDVI analysis were categorized by the four classes mentioned in Table 2. This study classified vegetation coverage under three different classes (sparse vegetation, medium vegetation, and dense vegetation). The rest of the values fell under the 'Others' class. After the raster images were obtained, the area covered by specific land coverage was calculated.

**Table 2.** Classification of the various types of vegetation in the study area, illustrating the color scheme used for the map.

| Type of Land Coverage | NDVI Value Range | Description | Corresponding Colors for Specific Classes |
|---|---|---|---|
| **Dense Vegetation** | 0.5–1 | The land is almost entirely covered by large plants, forests, trees, shrubs, herbs, etc. | |
| **Medium Vegetation** | 0.35–0.5 | Much of the land is covered by large plants, forests, trees, shrubs, herbs, etc. with some non-vegetated areas. Though NDVI values ranging in between 0.5 to 0.6 are usually categorized under medium vegetation coverage, many other studies have been found to categorize medium vegetation from 0.35 or 0.40 NDVI values. | |
| **Sparse Vegetation** | 0.2–0.35 | Only a small portion of the land is covered by large plants, forests, trees, shrubs, herbs, forest patches, etc. with a lot of non-vegetated areas. In most NDVI studies, sparse vegetation is detected when the NDVI value is around 0.2. | |
| **Others** | −1–0.2 | Land covered by water bodies, or transformed to agricultural lands or built area (comprising infrastructures like buildings, roads, industries, slums, artificial constructions, etc.), or existing as abandoned barren lands (such as landfills, exposed soil, empty land without any plants, etc.). | |

### 2.4.3. Accuracy Assessment

Google Earth Pro version 7.1.5.1557 and ERDAS IMAGINE Software were used for accuracy assessment. Only two classes of land use (Vegetation Coverage, and Non-Vegetation Coverage) were evaluated for accuracy.

## 3. Results

The results obtained from the spatiotemporal analysis of the vegetation coverage of the SBGE and their corresponding accuracy assessment are presented in the following subsections. The section concludes with a brief discussion on the limitations of the study.

### 3.1. Vegetation Cover Change of SBGE from 1995 to 2020

Figure 3 shows the vegetation coverage changes of the entire study area. It is observable that there was a spur of dense and medium vegetation from 1995 to 2000, the year when the SBGE was established. According to Figure 4 below (and Tables A1 and A2 in Appendix A), during this period, dense vegetation expanded from 0.068 km$^2$ to 7.56 km$^2$, an increase of 21.8%, while medium vegetation increased from 7.8 km$^2$ to 17.31 km$^2$, which was an increase of 27%. However, between 2000 and 2015, the area occupied by dense and medium vegetation decreased. In the year 2015, there was no dense or medium vegetation; vegetation was mostly sparse and much of the area (41.3%) was covered by other land uses. Nevertheless, vegetation coverage increased again between the years 2015 and 2020, wherein sparse vegetation replaced the other kinds of land use and some medium vegetation was also observed. During this period, sparse vegetation increased from 20.24 km$^2$ to 26.06 km$^2$ (16.9%) while medium vegetation had regenerated to cover 17.7% of the land area.

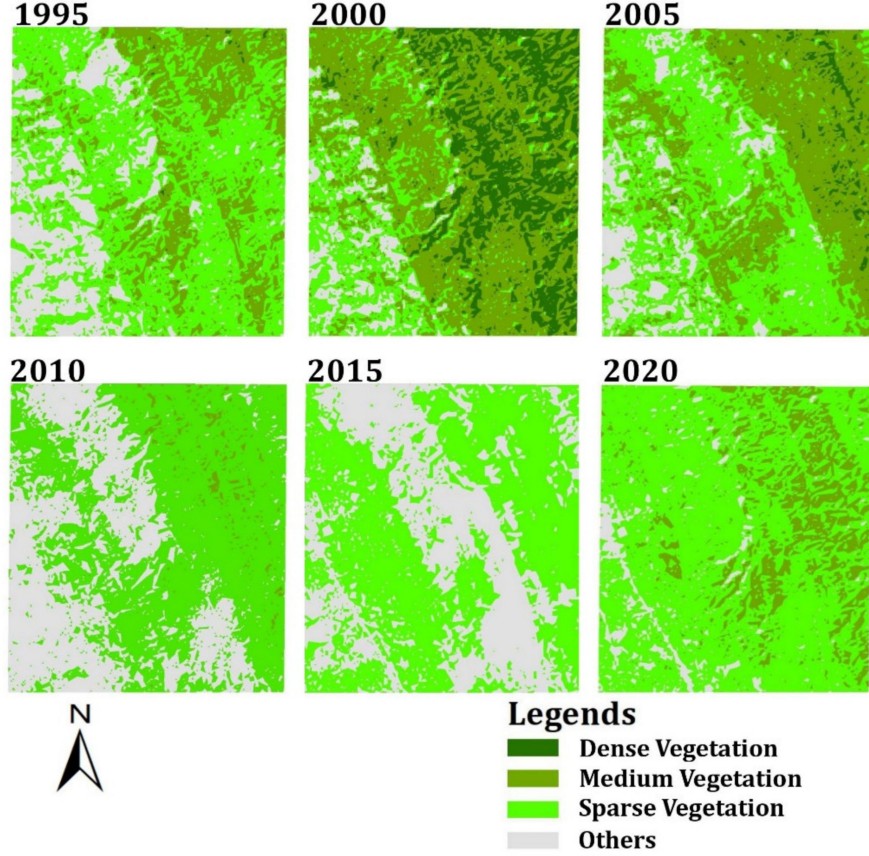

**Figure 3.** Vegetation coverage of the total area (1995–2020).

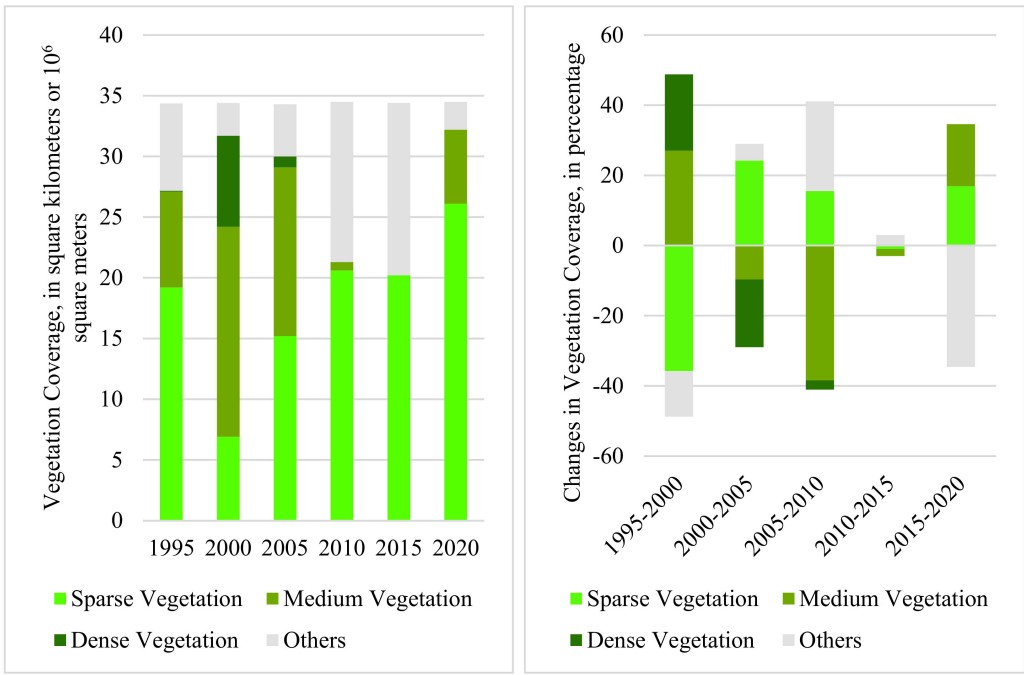

**Figure 4.** Vegetation coverage of the full study area (1995–2020).

For the SBGE area, according to Figures 5 and 6 (and Tables A3 and A4 in Appendix A), dense vegetation regenerated during the period of 1995 to 2000 to cover 1.73 km$^2$, or 18.6% of the land area, while medium vegetation increased from 2.19 km$^2$ to 5.74 km$^2$, an increase of 38.3%. During these years, sparse vegetation converted to dense and medium vegetation and thus covered only 1.51 km$^2$ (16.2%) by 2000. Following similar trends as the total study area, dense and medium vegetation kept decreasing from 2000, until they were completely removed by the year 2015. Much of the dense and medium vegetation was converted to sparse vegetation, which increased to cover 4.59 km$^2$ (49.4%) of the land, while the remaining were non-vegetated lands. Nevertheless, between 2015 and 2020, sparse vegetation increased to 7.43 km$^2$, a 30.6% increase, and was covering 80% of the land. Some medium vegetation also regenerated, covering 15.7% of the land by 2020.

For the surrounding area, according to Figures 7 and 8 below (and Tables A5 and A6 in Appendix A), dense vegetation regenerated during the period of 1995 to 2000 to cover 5.83 km$^2$, or 23.1% of the land area while medium vegetation increased from 5.79 km$^2$ to 11.56 km$^2$. The increase was 22.9% for both kinds of vegetation and covered a larger area than the SBGE. During these years, sparse vegetation was converted to dense and medium vegetation and thus covered only 5.38 km$^2$ (21.3%) by 2000. Following similar trends as the total study area and the SBGE, dense and medium vegetation kept decreasing from 2000, until they were removed by 2015. Much of the dense and medium vegetation was converted to sparse vegetation, which increased to cover 15.65 km$^2$ (62.1%) of the land, which meant the surrounding area possessed more vegetation than SBGE. Nevertheless, between 2015 and 2020, sparse vegetation increased to 18.63 km$^2$, an 11.8% increase, and was covering 73.9% of the land. Medium vegetation also regenerated and became prominent, covering 18.5% of the surrounding land by 2020.

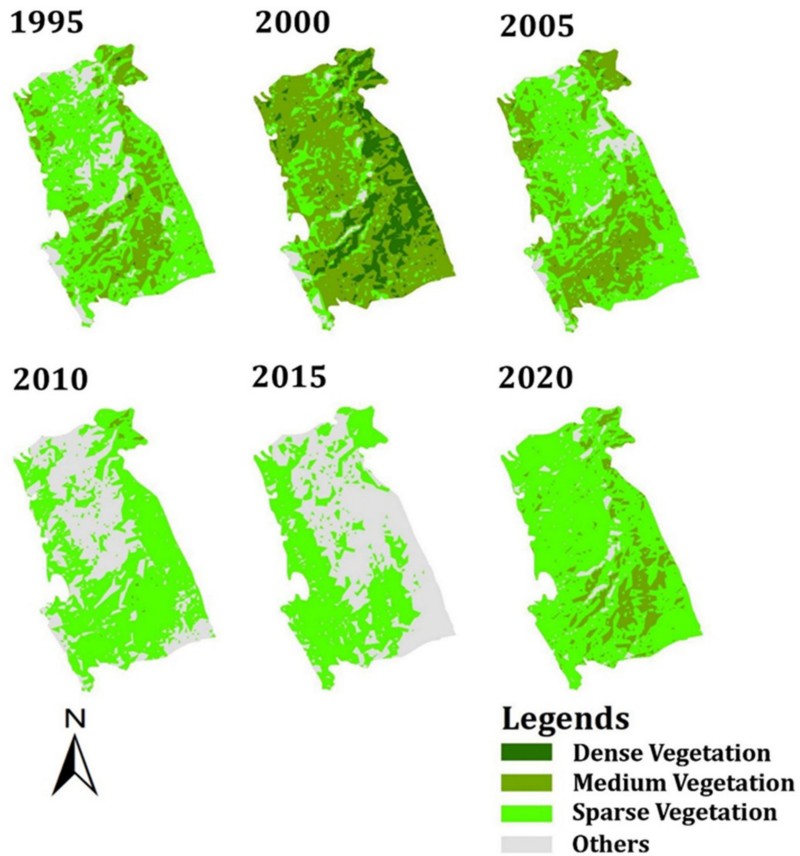

**Figure 5.** Vegetation coverage of the SBGE (1995–2020).

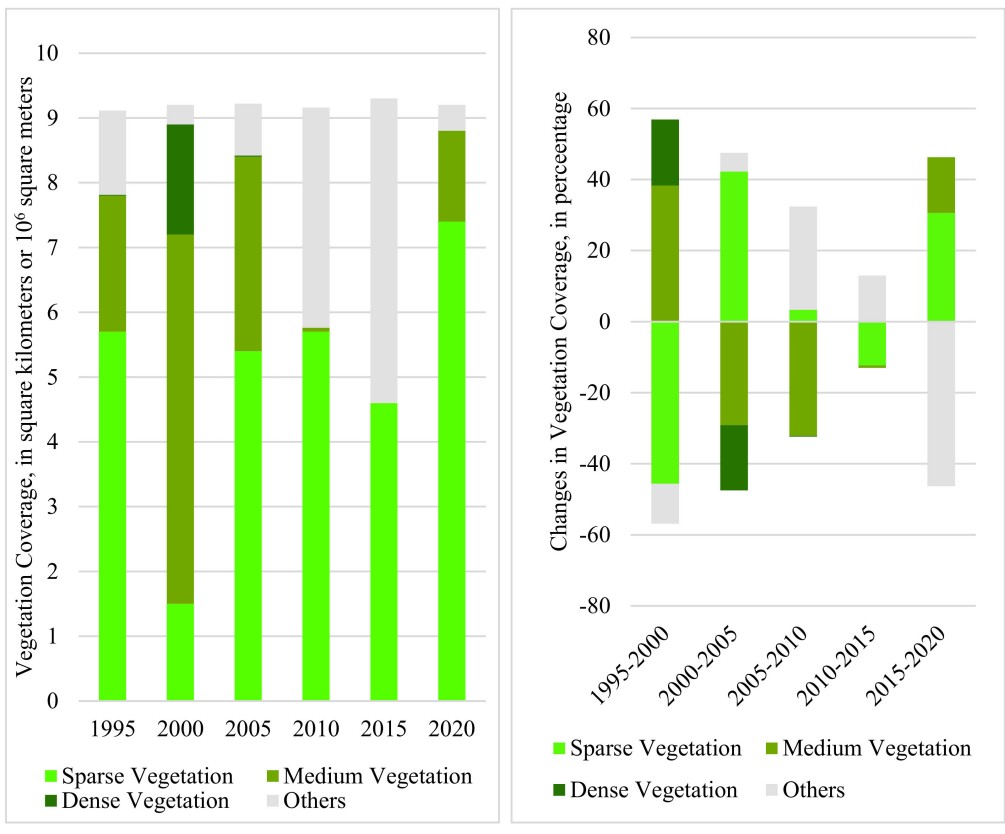

**Figure 6.** Vegetation coverage of the SBGE area (1995–2020).

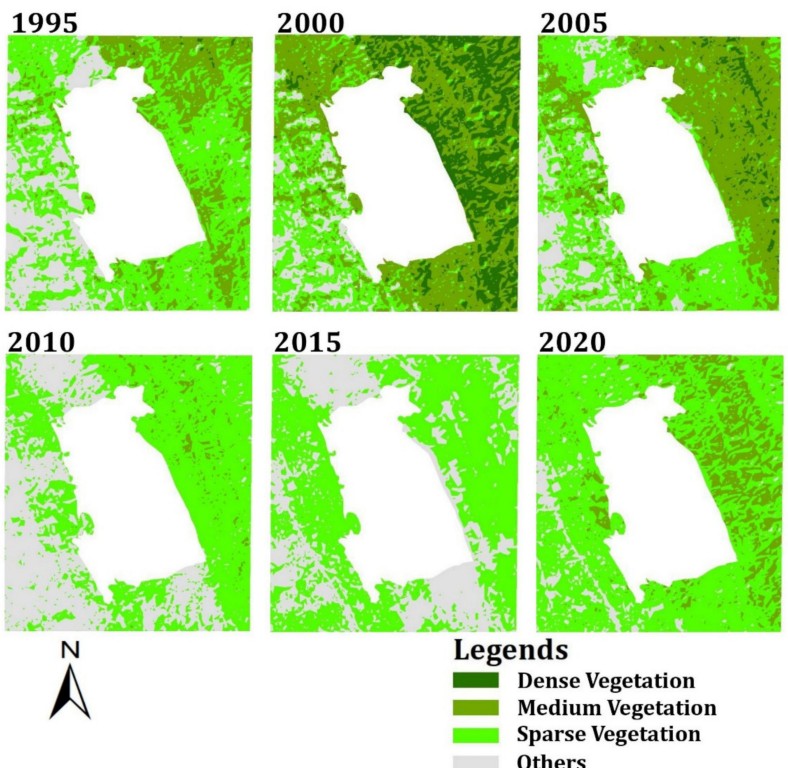

**Figure 7.** Vegetation coverage of the surrounding area (1995–2020).

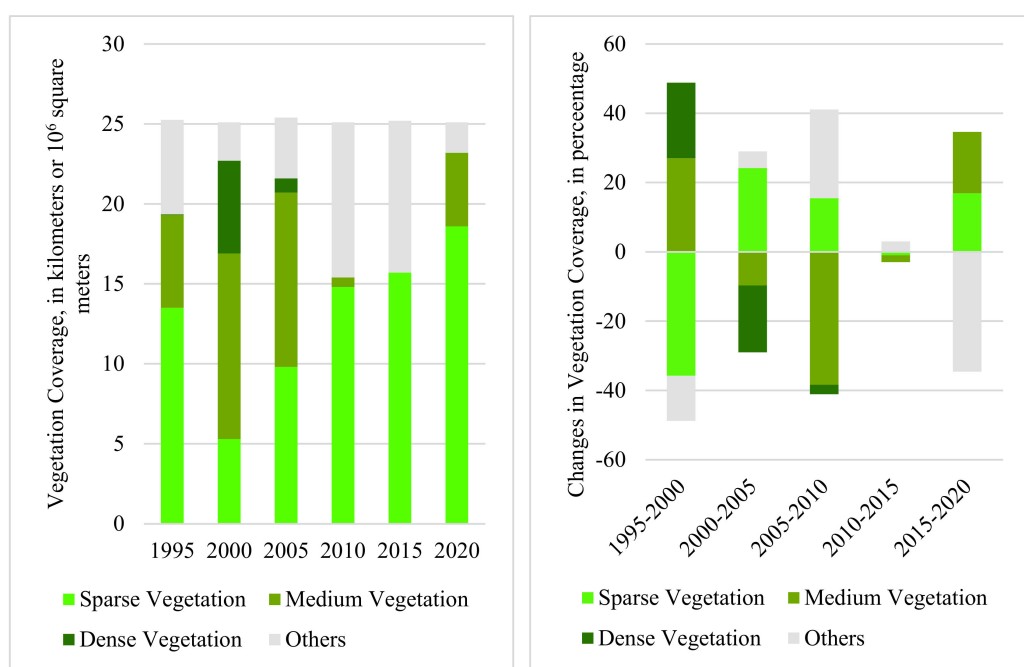

**Figure 8.** Vegetation Coverage of the Surrounding Area (1995–2020).

*3.2. Accuracy Assessment*

Table 3 displays the accuracy assessment results of the classified image. The classification of all of the years exhibited an accuracy of above 85%. While most years had an accuracy of 92% or above, the year 2015 had a slightly lower accuracy, which was 86%. The average accuracy of the classifications was 92.33%.

**Table 3.** Accuracy assessment data for the classified images of 1995, 2000, 2005, 2010, 2015, and 2020, in terms of the total area.

| Year | Land Coverage Type | Reference Totals | Classified Totals | Number Correct | Producer's Accuracy (in Percentage) | User's Accuracy (in Percentage) |
|---|---|---|---|---|---|---|
| | **Accuracy Totals** | | | | | |
| **1995** | *Vegetation* | 68 | 71 | 65 | 95.59 | 91.55 |
| | *Non-vegetation* | 32 | 29 | 29 | 90.625 | 100 |
| | *Totals* | 100 | 100 | 94 | | |
| | *Overall Classification Accuracy = 94%* | | | | | |
| **2000** | *Vegetation* | 75 | 75 | 71 | 94.67 | 94.67 |
| | *Non-vegetation* | 25 | 25 | 25 | 100 | 100 |
| | *Totals* | 100 | 100 | 96 | | |
| | *Overall Classification Accuracy = 96%* | | | | | |
| **2005** | *Vegetation* | 71 | 74 | 70 | 98.59 | 94.59 |
| | *Non-vegetation* | 29 | 26 | 23 | 79.31 | 88.46 |
| | *Totals* | 100 | 100 | 93 | | |
| | *Overall Classification Accuracy = 93%* | | | | | |
| **2010** | *Vegetation* | 81 | 78 | 75 | 92.59 | 96.15 |
| | *Non-vegetation* | 19 | 22 | 18 | 94.74 | 81.82 |
| | *Totals* | 100 | 100 | 93 | | |
| | *Overall Classification Accuracy = 93%* | | | | | |
| **2015** | *Vegetation* | 54 | 64 | 53 | 98.15 | 82.81 |
| | *Non-vegetation* | 46 | 34 | 33 | 71.74 | 97.01 |
| | *Totals* | 100 | 100 | 86 | | |
| | *Overall Classification Accuracy = 86%* | | | | | |
| **2020** | *Vegetation* | 73 | 72 | 70 | 95.89 | 97.22 |
| | *Non-vegetation* | 27 | 28 | 22 | 81.48 | 78.57 |
| | *Totals* | 100 | 100 | 92 | | |
| | *Overall Classification Accuracy = 92%* | | | | | |
| **AVERAGE ACCURACY = 92.33%** | | | | | | |

### 3.3. Limitations

There are some major limitations of this study. Firstly, the satellite imageries had a resolution of only 30 m, which means if any significant class was present within 30 m × 30 m, it would not be detectable, and thus may have been misclassified. Satellite images of better resolution may be available, but they are usually not provided in open sources such as in GloVis. Secondly, the advantage of assessment of vegetation coverage using NDVI is generally limited to any possible linearity of its functional relationship with vegetation properties (e.g., biomass). The NDVI product also carries only some of the information available in the original spectral reflectance data. Moreover, the calculation of the NDVI value is sensitive to several perturbing factors, such as cloud cover, atmospheric effects, soil effects, etc. Thirdly, even though the accuracy of 85% is generally acceptable in GIS-RS analysis [68], the accuracy of classification below 100% in this study implies that some pixels were misclassified, which could have been avoided with more precise analysis. In addition, the misclassified areas could have been corrected to proper classes through manual editing. Attempts were made to mask the cloud coverage in the satellite imageries,

which was particularly high for the satellite imagery used in 1995. While a cloud coverage of less than 20% is acceptable, this may have introduced inaccuracies in the GIS-RS analysis. Moreover, accuracy assessment was limited to the analysis of only two classes of land use, vegetation and the lack thereof, which may have contributed to imprecise accuracy percentages. Collecting ground truth information may have mitigated some of the limitations pertaining to accuracy, especially the verification of coordinates using GPS [74] as well as the information stored in "pixels" in comparison with reality [75], but travel restrictions imposed due to the COVID-19 pandemic did not allow for such validation measures.

## 4. Discussion and Recommendations

Data analysis revealed that vegetation coverage significantly increased from 1995 to 2000. Vegetation growth was perceived in 84.6% of the total area during this period, of which 21.8% was covered by dense forest. However, the area of dense vegetation was higher in the surrounding area (22.9%) than the SBGE itself (18.6%), while the area covered by medium vegetation was higher in SBGE (38.3%) than the surrounding areas (22.9%). This progress could denote that the intensive forest management practices in the SBGE proved to be a success, which translated to the surrounding areas as well. Even though the SBGE was only established in 2000, because the entire project proceedings began a few years prior, the improvements in vegetation coverage could be perceived at least as early as 2000.

However, unlike the anticipated outcome of the project and findings from previous studies [48], spatiotemporal analysis revealed, with over 85% accuracy, a trend of decreasing vegetation coverage over the years. The vegetation coverage of the entire study area decreased from 2000 to 2015, where 33.4% of vegetation was lost to other forms of non-vegetation land uses. Much of the dense and medium vegetation was either converted to sparse vegetation or removed to create space for other land uses. Even as early as 2005, nearly half of the SBGE was covered by sparse vegetation, although the SBGE itself had higher vegetation coverage (91.5%) than the nearby areas (85.8%). The year 2015 experienced the worst scenario in terms of vegetation coverage. Nearly half (41.3%) of the entire study area was composed of non-vegetated land coverage and the remaining areas had little to no medium or dense vegetation. The total vegetation coverage of the SBGE in 2015 was found to be lower (49.4%) than in the nearby areas (62.1%).

Incidences of lower vegetation cover in SBGE could be a result of the activities of individuals from local communities who rely on forest resources for their livelihoods. The local community used to largely depend on the SBGE, which was a good source of income for them [45]. Currently, however, their use of forest resources has been capped due to protectionary measures. Locals can only collect non-timber wood products, non-wood forest products, and fruit and medicinal plants [45,48]. Prohibition on the extraction of forest products from the SBGE and restriction on the use of park roads to enter adjacent forests make the livelihoods of surrounding villagers vulnerable, which has led to poor perception of the intentions of conservation practices among locals [45,48]. As a result, some of the locals engage in illegal activities such as firewood and exotic plant collection, grazing in the park area, and poaching wildlife [45,48]. Frequent fires are set by the locals to promote the growth of the invasive sun grass (*Imperata cylindrica*), which is harvested for their livelihood, roofing, fodder, and cultural needs [43,76]. As discussed, deforestation rates have been observed to be higher within protected areas in other ecotourism spots around the world as well, such as the Wolong Giant Panda Nature Reserve [65]. The case study also singled out disturbances by the local community within the Reserve as the primary cause of the destruction of the forest and panda habitat.

The conflict between the ambitions of the SBGE and the lifestyles of locals is attributed to the fact the very few local people were involved during the planning, establishment, or operation of the SBGE [48,77]. Even though the park establishment has created some job opportunities, very few local households are able to participate in such opportunities [45,78], so the economic benefit for the locals has been very low [48]. There has, however, been an improvement in the awareness of the importance of forests and tourism

in people, as well as in communication, connection, and cultural exchange [48]. A willingness to participate in co-management practices and tourism business was perceived among the locals [48]. Therefore, the absence of co-management practices in the SBGE [48,77] can be addressed through a partnership between local communities residing near protected areas and governmental actors, which is crucial in reaching conservation targets [79]. The importance of collaboration with the local community and other stakeholders has been recognized in other studies as well, such as in the Gunung Leuser National Park of Indonesia by Hartoyo et al. [80], in order to retain its densely vegetated state. In a recent study [81] conducted in Chambok, Cambodia, community-based ecotourism was found to effectively lower deforestation, revealed through both satellite imagery analysis and surveys. Conservation can only be successful when there is a sense of ownership of the area, validation through the decision-making for the protected area, and the ability to receive benefits from the area [82]. Thus, collaboration and consideration of resource users and resource regulators are important for successful ecological restoration [83].

The decreasing plant diversity over the years was evident in the satellite imageries. Urbanization can affect plant density and diversity by restricting the space for vegetation growth and reducing soil moisture and nutrient contents [84]. Numerous spatiotemporal analysis studies of land use and land cover changes (LULCC) have been conducted in Bangladesh that have attributed unplanned urbanization to the decreasing vegetation coverage [68,85]. Most of these studies found high rates of transformation of the built-up area from fallow lands, hills, and vegetation over the years. Similar to this study, Nath [86] also conducted an NDVI analysis of the Bandarban Hill Tracts and found that forest cover decreased by 15.47% from 1989 to 2010 due to natural and different anthropogenic activities.

While direct anthropogenic disturbances by visitors have been minimal [48], development projects for tourism contribute to a significant amount of disturbances, where soil diggings due to construction work have been found throughout the SBGE [48]. One study [87] conducted in the SBGE found that species richness and basal area are negatively related to anthropogenic disturbances. In addition, the construction of the built area around the study area possibly contributed to reduced soil fertility in the study area. Growing population and, subsequently, socio-economic needs tend to lead to the conversion of forested lands to urban settlements in an unplanned way [67]. The soil quality of forest lands may be adversely affected by deforestation, soil erosion, shifting cultivation, and reduced fallow periods due to increased population pressure [88]. In fact, human activities have been consistently singled out as hindering conservation in the SBGE, which is resulting in less species diversity [47,87,89,90].

Nevertheless, remarkably, the satellite imageries of 2020 exhibited the second largest vegetation growth in the six years that were analyzed. Vegetation covered 93.3% of the total study area, of which 17.7% were medium vegetation. This growth was an increase of 34.6% compared with 2015. However, the SBGE, yet again, displayed less coverage of medium vegetation (15.7%) compared with the surrounding areas (18.5%). Despite the higher prevalence of sparse vegetation, the overall vegetation coverage was higher in the SBGE than in the surrounding areas. One reason for this regrowth observed could be that urbanization has slowed down and decreased over the years. In fact, the period of 2010–2020 saw a decreased land area of non-vegetation, from 38.3% of the entire study area in 2010 to a mere 6.7% in 2020. In SBGE, only 4.3% of the area was the non-vegetation land coverage type in 2020. Since the expansion of the built area slowed, the adverse impacts on the soil quality and vegetation were potentially naturally attenuated, leading to the healthy growth of vegetation in the study area. Current decelerating rates of vegetated land conversion need to persist to reverse the impacts of anthropogenic activities and make progress towards the goals of the SBGE project. Another likely reason for this increase in vegetation could be attributed to the 'U' shape relationship between the changes in NDVI in urbanized areas of cities and rates of urbanization, as found by Du et al. [91] in a study conducted in China. Du et al. [91] revealed that with the continuous progress of urbanization level, the adverse effect of urbanization on vegetation gradually diminished

or even disappeared over the last four decades. To understand whether this 'U' shape pattern is also manifesting in the context of the study area, future research spanning a larger time period, at least as much as four decades, is required. This is because contrary to the study by Du et al. [91], Yao et al. [92] assessed the impact of urbanization on vegetation for 59 cities in Africa from 2001 to 2017 and found that, on average, 60% of the urban areas displayed significant decreasing of annual changes in enhanced vegetation index.

Even though unsustainable forms of urbanization and interventions from the local communities have been suggested as the major contributor behind the trend in vegetation coverage of the SBGE observed, the uncertainty causes a gap to remain regarding the main cause behind the trend. Therefore, future research work needs to be directed towards understanding the primary socio-economic data and drivers of change. For instance, models incorporating elevation, climate data, and the income derived from tourism, and considerations of the changing political economy are some aspects that need further exploring. Moreover, remote-sensing data with a higher spatial resolution are needed to detect smaller changes in the SBGE across shorter time intervals. More importantly, future ecotourism initiatives should integrate GIS-RS technologies in the decision-making processes, so that sensitive ecological areas can be properly delineated from urban settlements with adequate regions of buffer zones.

Bangladesh, possessing a transition economy, is undergoing rapid urbanization. Hasan et al. [93] predicted that the economic growth scenario would have profound effects on the CHT due to an increasingly higher share of built area. Nevertheless, if vegetation restoration efforts were integrated with increasing demand for a high-quality urban environment, the urbanization process may not necessarily lead to vegetation degradation on a large scale [94]. This is why the sustainable use of land with the consideration of proper resource management is important for the sustainable development of Bangladesh. Sustainable land-use policies, therefore, will play a key role in Bangladesh towards ensuring the development and use of land resources that simultaneously improve people's living standards [95] and are compatible with the carrying capacity of the environment [96]. Before the SBGE faces the same fate as the Himchari National Park of degradation and forest fragmentation [67], municipal authorities of the Sitakunda Upazila need to act rapidly to save and protect the SBGE.

In spite of its limitations, this study was able to effectively portray the inadequacies in current ecotourism initiatives in Bangladesh and echoed some of the major failures of ecotourism interventions in many of the developing nations. Consequently, similar policy implications, i.e., of community-based ecotourism and co-management practices, and sustainable land-use policies, should be translated for nations grappling with the same issue of declining forest coverage resulting from ecotourism interventions.

## 5. Conclusions

The SBGE, one of the first ecoparks and ecotourism spots in Bangladesh, hosts a vast range of biodiversity, with hundreds of species of flora and fauna. While the anticipated outcome of the project was to preserve and rejuvenate indigenous species, satellite imagery analysis of vegetation coverage via GIS-RS revealed that despite a brief increase in vegetation coverage of 84.6% from 1995 to 2000, the year when the SBGE was established, the vegetation coverage fell drastically from 2000 to 2015, wherein 33.4% of vegetation was lost, and much of the dense and medium vegetation was either been converted to sparse vegetation or other forms of land uses. The most likely reason for the great decline in vegetation is suspected to be anthropogenic activities, namely, unplanned urbanization. Surprisingly enough, vegetation was higher in the surrounding areas of the SBGE compared with the SBGE itself, despite the park's protectionary status. Incidences of lower vegetation cover in SBGE could be a result of the activities of individuals from local communities who rely on forest resources for their livelihoods, wherein the conflicting ambitions of the locals and the SBGE project have been affecting the success of the conservation initiatives due to a lack of participation of the locals. From the period of 2015 to 2020, however, vegetation was seen

to regenerate, potentially due to the decelerating urbanization or the possible manifestation of the 'U' shape relationship between the changes in NDVI in urbanized areas of cities and rates of urbanization. Sustainable land-use policies, incorporating community-based ecotourism and co-management practices, may help attain the targets of the project and lead the SBGE to emerge as a success story of the Bangladeshi ecotourism industry.

**Author Contributions:** Conceptualization, N.R., S.M.N.U. and M.G.; methodology, N.R. and S.N.; validation, S.N. and R.R.; formal analysis, S.N.; investigation, N.R. and S.N.; resources, S.M.N.U. and M.G.; data curation, N.R. and S.N.; writing—original draft preparation, N.R.; writing—review and editing, S.M.N.U. and M.G.; visualization, S.N. and R.R.; supervision, S.M.N.U. and M.G.; project administration, N.R. and S.N.; funding acquisition, S.M.N.U. All authors have read and agreed to the published version of the manuscript.

**Funding:** This research received no external funding.

**Data Availability Statement:** Satellite imageries were collected from the publicly available USGS GloVis, which can be found here: https://glovis.usgs.gov/, (accessed on 2 October 2021). All of the analyzed data are provided within the main text of this article and the Appendix A.

**Conflicts of Interest:** The authors declare no conflict of interest.

## Appendix A

**Table A1.** Vegetation Coverage of the Full Study Area.

| | TOTAL AREA (Surrounding + Botanical Garden) | | | | | | | | | | | |
| | 1995 | | 2000 | | 2005 | | 2010 | | 2015 | | 2020 | |
| | Square Meters | % of Land | Square Meters | % of Land | Square Meters | % of Land | Square Meters | % of Land | Square Meters | % of Land | Square Meters | % of Land |
|---|---|---|---|---|---|---|---|---|---|---|---|---|
| Sparse Vegetation | 19,239,308 | 55.8 | 6,888,217 | 20.0 | 15,232,752 | 44.2 | 20,602,421 | 59.7 | 20,242,389 | 58.7 | 26,058,632 | 75.6 |
| Medium Vegetation | 7,979,732 | 23.1 | 17,309,269 | 50.1 | 13,944,091 | 40.4 | 679,920.6 | 2.0 | 616.2223 | negligible | 6,111,398 | 17.7 |
| Dense Vegetation | 68,642.77 | 0.2 | 7,562,188 | 22.0 | 945,232.3 | 2.7 | 0 | 0 | 0 | 0 | 0 | 0 |
| Others | 7,201,843 | 20.9 | 2,729,853 | 7.9 | 4,367,451 | 12.7 | 13,207,185 | 38.3 | 14,246,522 | 41.3 | 2,319,496 | 6.7 |
| Total | 34,489,527 | 100 | 34,489,527 | 100 | 34,489,527 | 100 | 34,489,527 | 100 | 34,489,527 | 100 | 34,489,527 | 100 |

**Table A2.** Changes in Vegetation Coverage of the Full Study Area.

| | TOTAL AREA (Surrounding + Botanical Garden) | | | | | | | | | |
| | 1995–2000 | | 2000–2005 | | 2005–2010 | | 2010–2015 | | 2015–2020 | |
| | Square Meters | % of Land | Square Meters | % of Land | Square Meters | % of Land | Square Meters | % of Land | Square Meters | % of Land |
|---|---|---|---|---|---|---|---|---|---|---|
| Sparse Vegetation | −12,351,091 | −35.8 | +14,544,535 | +24.2 | +5,369,669 | +15.5 | −360,032 | −1.0 | +5,816,243 | +16.9 |
| Medium Vegetation | +9,329,537 | +27 | −3,365,178 | −9.7 | −13,264,170 | −38.4 | −679,274 | ~−2.0 | +6,110,782 | +17.7 |
| Dense Vegetation | +7,493,545 | +21.8 | −6,616,956 | −19.3 | −945,232 | −2.7 | 0 | 0 | 0 | 0 |
| Others | −4,471,990 | −13 | +1,637,598 | +4.8 | +8,839,734 | +25.6 | +1,039,337 | +3.0 | −11,927,026 | −34.6 |

**Table A3.** Vegetation Coverage of the SBGE Area.

| | SBGE AREA | | | | | | | | | | | |
|---|---|---|---|---|---|---|---|---|---|---|---|---|
| | 1995 | | 2000 | | 2005 | | 2010 | | 2015 | | 2020 | |
| | Square Meters | % of Land | Square Meters | % of Land | Square Meters | % of Land | Square Meters | % of Land | Square Meters | % of Land | Square Meters | % of Land |
| **Sparse Vegetation** | 5,733,777 | 61.8 | 1,506,777 | 16.2 | 5,423,909 | 58.4 | 5,731,433 | 61.7 | 4,588,935 | 49.4 | 7,427,325 | 80.0 |
| **Medium Vegetation** | 2,187,867 | 23.6 | 5,746,309 | 61.9 | 3,048,513 | 32.8 | 57,743.09 | 0.7 | 0 | 0 | 1,458,786 | 15.7 |
| **Dense Vegetation** | 14,220.97 | 0.1 | 1,733,198 | 18.7 | 21,501.56 | 0.3 | 0 | 0 | 0 | 0 | 0 | 0 |
| **Others** | 1,345,882 | 14.5 | 295,463.3 | 3.2 | 787,824.1 | 8.5 | 3,492,571 | 37.6 | 4,692,812 | 50.6 | 395,635.6 | 4.3 |
| **Total** | 9,281,747 | 100 | 9,281,747 | 100 | 9,281,747 | 100 | 9,281,747 | 100 | 9,281,747 | 100 | 9,281,747 | 100 |

**Table A4.** Changes in the Vegetation Coverage of the SBGE Area.

| | SBGE AREA | | | | | | | | | |
|---|---|---|---|---|---|---|---|---|---|---|
| | 1995–2000 | | 2000–2005 | | 2005–2010 | | 2010–2015 | | 2015–2020 | |
| | Square Meters | % of Land | Square Meters | % of Land | Square Meters | % of Land | Square Meters | % of Land | Square Meters | % of Land |
| **Sparse Vegetation** | −4,227,000 | −45.6 | +3,917,132 | +42.2 | +307,524 | +3.3 | −1,142,498 | −12.3 | +2,838,390 | +30.6 |
| **Medium Vegetation** | +3,558,442 | +38.3 | −2,697,796 | −29.1 | −2,990,770 | −32.1 | −57,743 | −0.7 | +1,458,786 | +15.7 |
| **Dense Vegetation** | +1,718,977 | +18.6 | −1,711,696 | −18.4 | −21,501 | −0.3 | 0 | 0 | 0 | 0 |
| **Others** | −1,050,419 | −11.3 | +492,361 | +5.3 | +2,704,747 | +29.1 | +1,200,241 | +13.0 | −4,297,176 | −46.3 |

**Table A5.** Vegetation Coverage of the Surrounding Area.

| | SURROUNDING AREA | | | | | | | | | | | |
|---|---|---|---|---|---|---|---|---|---|---|---|---|
| | 1995 | | 2000 | | 2005 | | 2010 | | 2015 | | 2020 | |
| | Square Meters | % of Land | Square Meters | % of Land | Square Meters | % of Land | Square Meters | % of Land | Square Meters | % of Land | Square Meters | % of Land |
| **Sparse Vegetation** | 13,505,531 | 53.6 | 5,381,440 | 21.3 | 9,808,844 | 38.9 | 14,870,989 | 59.0 | 15,653,454 | 62.1 | 18,631,308 | 73.9 |
| **Medium Vegetation** | 5,791,865 | 23.0 | 11,562,960 | 45.9 | 10,895,578 | 43.2 | 622,177.5 | 2.5 | 616.2223 | ~ 0 | 4,652,611 | 18.5 |
| **Dense Vegetation** | 54,421.8 | 0.2 | 5,828,990 | 23.1 | 923,730.7 | 3.7 | 0 | 0 | 0 | 0 | 0 | 0 |
| **Others** | 5,855,962 | 23.2 | 2,434,390 | 9.7 | 3,579,627 | 14.2 | 9,714,614 | 38.5 | 9,553,710 | 37.9 | 1,923,861 | 7.6 |
| **Total** | 25,207,780 | 100 | 25,207,780 | 100 | 25,207,780 | 100 | 25,207,780 | 100 | 25,207,780 | 100 | 25,207,780 | 100 |

**Table A6.** Changes in the Vegetation Coverage of the Surrounding Area.

| | SURROUNDING AREA | | | | | | | | | |
| --- | --- | --- | --- | --- | --- | --- | --- | --- | --- | --- |
| | 1995–2000 | | 2000–2005 | | 2005–2010 | | 2010–2015 | | 2015–2020 | |
| | Square Meters | % of Land | Square Meters | % of Land | Square Meters | % of Land | Square Meters | % of Land | Square Meters | % of Land |
| **Sparse Vegetation** | −8,124,091 | −32.3 | +4,427,404 | +17.6 | +5,062,145 | +20.1 | +782,465 | +3.1 | +2,977,854 | +11.8 |
| **Medium Vegetation** | +5,771,095 | +22.9 | −667,382 | −2.7 | −10,273,400 | −40.7 | −621,561 | −2.5 | +4,651,995 | +18.5 |
| **Dense Vegetation** | +5,774,568 | +22.9 | −4,905,259 | −19.4 | −923,730 | −3.7 | 0 | 0 | 0 | 0 |
| **Others** | −3,421,572 | −13.5 | +1,145,237 | +4.5 | +6,134,987 | +24.3 | −160,904 | −0.6 | −7,629,849 | −30.3 |

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
