# Peer review of "Ecotourism as a Forest Conservation Tool: An NDVI Analysis of the Sitakunda Botanical Garden and Ecopark in Chattogram, Bangladesh"

_sustainability, doi:10.3390/su132112190_

Round 1

Reviewer 1 Report

Dear Authors,

thanks for allowing me to review your manuscript on ecotourism in a botanical garden and ecopark in Bangladesh. I have a few concerns about this manuscript:

  • the most relevant part of the manuscript is the "Discussion and recommendation" sections which is very interesting. However, this section contains new information and literature and it fails to discuss your results. It seems that this part is completely different from the research you did. In the methods section you describe how you collected and analysed satellit images, but you do not mention any other source/data used to support your comment that the changes in vegeatation are caused by locals setting fire to different places.
  • I suggest restructuring the paper and to link better the current "discussion and recommendation" section to the methods and teh results part. 
  • Parts of the conclusions could go to the introduction. In the current version, the introduction is too vague.
  • I struggle to find data on ecotourism. How do you link your research with satellit images on various types of vegetation and ecotourism?
  • This manuscript is more on forest preservation than on ecotourism. The discussion section is built on suppositions or maybe another research, but surely it is not a discussion of the data gathered and analysed in this manuscript.

Author Response

REVIEWER 1

Comment: Dear Authors,

thanks for allowing me to review your manuscript on ecotourism in a botanical garden and ecopark in Bangladesh. I have a few concerns about this manuscript:

Response: We thank the reviewer for taking the time to thoroughly review our manuscript and offer their very useful feedback in helping us improve the quality of the manuscript.

Comment: the most relevant part of the manuscript is the "Discussion and recommendation" sections which is very interesting. However, this section contains new information and literature and it fails to discuss your results. It seems that this part is completely different from the research you did. In the methods section you describe how you collected and analysed satellit images, but you do not mention any other source/data used to support your comment that the changes in vegeatation are caused by locals setting fire to different places.

I suggest restructuring the paper and to link better the current "discussion and recommendation" section to the methods and teh results part. 

Response: We appreciate and understand the concern of the reviewer here. However, what we attempted to do here was suggest plausible reasons for the vegetation coverage change in the SBGE area. Such vegetation coverage changes are sometimes complemented with some socio-economic data or social science methods and regression models to support the association between tourism and vegetation coverage change. However, as this study was conducted when the pandemic situation had been serious, with high rates of death and infectivity and intensified mobility restrictions (early 2020-early 2021), no primary data could be collected to complement the findings from the GIS-RS analysis. Therefore, we resorted to comprehensively review all the possible literature that we could find on the SBGE and the socio-economic activities of the local community. We rephrased our objectives as follows:

“Similar studies, even though few in number, produced over the years that assess the impacts of ecotourism on forest protection in biodiversity hotspots tended to measure the forest coverage change before and after tourism implementation [13], often complemented with some socio-economic data or social science methods and regression models to support the association. The Wolong Giant Panda case study, for example, represents a clear understanding of how coupling between human and natural systems vary across spatial units. The understanding of the complexities between human and natural systems is often hindered by the academic separation of ecological and social sciences [66]. Therefore, an interdisciplinary perspective comprising ecological and social sciences are crucial to study coupled human and natural systems [66]. However, as this study was conducted when the pandemic situation had been serious, with high rates of death and infectivity and intensified mobility restrictions (early 2020-early 2021), no primary data could be collected to complement the findings from the GIS-RS analysis (more on Section 3.3. Limitations). Thus, this study attempts to suggest and conceptualize plausible anthropogenic activities that may have contributed to the changes in vegetation by providing insights into the tourism practices in and around the SBGE, and the socio-economic activities of the local community via a comprehensive literature review of relevant studies surrounding the SBGE.” (1.5. Objectives, paragraph 1, page 5).

The study by Liu et al. (2001), which undertook a study in the Wolong Nature Reserve for Giant Pandas, also found similar results as ours (i.e. higher deforestation rate within Reserves than outside). They employed a review of relevant literature to suggest their reasoning behind this deforestation trend, namely population growth and local interference.

“As discussed, deforestation rates have been observed to be higher within protected areas in other ecotourism spots around the world as well, such as the Wolong Giant Panda Nature Reserve [60]. The case study has also singled out disturbances by the local community within the Reserve as the primary cause of the destruction of the forest and panda habitat.” (4. Discussion and Recommendations, paragraph 3, page 16).

We also used a similar approach to provide suggestions for this decline in vegetation and came to a similar conclusion. Therefore, while we could not strongly conclude that local or tourism initiatives caused these trends, we have simply provided possible reasons for this. We felt engaging in such discussions would have better policy implications than simply stating the problem. The recommendations under this section also comprised new literature for similar reasons. Regardless, we have now identified this limitation as an avenue for future research.

“Even though unsustainable forms of urbanization and interventions from the local communities have been suggested as the major contributor behind the trend in vegetation coverage of the SBGE observed, the uncertainty causes a gap to still remain regarding the main cause behind the trend. Therefore, future research work needs to be directed towards understanding the primary socio-economic data and drivers of change. For instance, models incorporating elevation, climate data, and the income derived from tourism, and considerations of the changing political economy are some aspects that need further exploring. Moreover, remote sensing data with a higher spatial resolution are needed to detect smaller changes in the SBGE across shorter time intervals. More importantly, future ecotourism initiatives should integrate GIS-RS technologies in the decision-making processes, so that sensitive ecological areas can be properly delineated from urban settlements with adequate regions of buffer zones.” (4. Discussion and Recommendations, paragraph 8, page 17-18).

Comment: Parts of the conclusions could go to the introduction. In the current version, the introduction is too vague.

Response: The conclusion summarizes the major findings of our study and some of the key messages from our discussion. Therefore, it does not seem appropriate to shift parts of the conclusion to the introduction. The introduction was planned to be structured as follows: an overview of ecotourism and its contributions to forest conservation, a brief snapshot of Bangladesh’s natural resources and ecotourism spots, an introduction to the study area that is located in Bangladesh, contributions of GIS-RS (the study’s main method) in the field of ecotourism and forest conservation, and objectives. We believe that such structuring creates the foundation for the sections that follow the introduction. If there are specific changes that you would like to see, such as specific parts from the Conclusion shifted to the Introduction, or specific parts from the Introduction reworked, please do let us know. We will do our best to incorporate the changes.

Comment: I struggle to find data on ecotourism. How do you link your research with satellit images on various types of vegetation and ecotourism?

Response: It seems like the linkage between ecotourism and vegetation coverage had not been properly communicated in the initial draft. Often, ecotourism is implemented in sensitive ecological regions, many of which are forested lands. Overall, tourism has been found to affect the vegetation coverage of such spots, mostly negatively, as such interventions tend to increase the use of forest products, such as to build infrastructures. To illustrate this, studies (albeit very few in number; less than 20 since 2000) have been conducted to assess the impacts of ecotourism on forest protection in biodiversity hotspots by measuring the forest coverage change before and after tourism implementation. The same method has been adopted for this study. The link between ecotourism and vegetation cover has been further explained:

“Ecotourism practiced in areas with impressive biodiversity and landscapes is a promising sub-sector of tourism [11,12] but has often been criticized for being ineffective and/or harmful, initiating numerous environmental risks such as water pollution and old-growth deforestation due to the increasing reliance and usage of natural resources, especially forest products when the ecotourism spot is or nearby a forest [13–15]. In many case studies, activities related to tourism increased the demand for timber and fuelwood for the construction of new infrastructures, including housing [13]. The economic development of the area also acted as pull factors for migration and population growth, and many forested lands were cleared for other land uses [13]. However, most of the claims opposing ecotourism are attributed to flawed research designs when studying the topic of ecotourism, making it difficult to assess the simultaneous economic, environmental, and social benefits it offers [14]. On the contrary, ecotourism has been reported to lead to forest regeneration, particularly in agrarian landscapes, when approached with conservation mechanisms, such as protected areas, Payment for Ecosystem Services (PES), and monitoring/enforcement [13]. For instance, when a PES system was instituted at the Monarch Butterfly Biosphere Reserve in Mexico, more gains than losses were experienced in the closed cover density during 1999-2009 [16]. Therefore, ecotourism can be viewed as an incentive-driven forest governance intervention. In addition, by creating the perception of biodiversity as ‘economic goods’ [17], ecotourism can bring advantages in conservation by supporting wildlife and protected areas, diversifying livelihoods, promoting environmental interpretation and ethics, and strengthen resource management [14]. The income generated from ecotourism can be also used for the landscape-scale conservation of habitats for a diverse group of animals and plants [11,18]. Several ecotourism initiatives, such as the Chitwan National Park in Nepal [13,19], are illustrations of ecotourism as a promising forest conservation tool. In the Chitwan National Park, satellite image analysis displayed regeneration of many forest patches after the introduction of a buffer zones program due to significant investment in plantation and forest-management initiatives.” (1.1. Ecotourism as a Tool for Sustainable Development and Biodiversity Conservation, paragraph 3, page 2).

“The application of GIS has also been recommended in the assessment of biodiversity and forest conservation practices [57–59]. Overall, however, there is a dearth of literature that empirically analyzes ecotourism impacts on forests. Brandt and Buckley [13] found only 17 studies, published between 2000 to 2018, which evaluated the potential of ecotourism for forest protection in biodiversity hotspots. The majority of the studies (14 of 17) evaluated forest change using satellite data, while the remaining used methods from social sciences (such as surveys and interviews). The archetypal case study for the negative forest outcomes of ecotourism, for instance, was illustrated by Liu et al. [60] in the analysis of the impacts of tourism on the Wolong Giant Panda Nature Reserve in southwest China. The study had revealed that there was an increase in deforestation after the Reserve had been implemented, and found that, surprisingly, deforestation had occurred more within the Reserve than outside. Similar studies were conducted during the aforementioned period in other ecotourism spots in China, and other developing nations like India, Nepal, Cambodia, Mexico, Belize, and Peru [13]. The most recent literature [61], focused on the land use and cover of the Shivpuri watershed in Nepal, also repeat a similar scene- a 110 ha reduction in forest coverage between 1999 to 2016. However, the application of GIS and/or RS technology to assess the effectiveness of biodiversity conservation strategies, such as in protected areas, have been less extensive, even more so in Bangladesh. Nevertheless, one study found that the Himchari National Park, a protected area has been degraded, fragmented, and converted severely into various land uses, where nearly half of the dense forest land was converted to other land uses in the period of 1977 to 2017 [62].” (1.4. Application of GIS in Ecotourism Research and Vegetation Cover Analysis, paragraph 3, page 4)

Comment: This manuscript is more on forest preservation than on ecotourism. The discussion section is built on suppositions or maybe another research, but surely it is not a discussion of the data gathered and analysed in this manuscript

Response: Thank you for your feedback. We believe your feedback has allowed our paper to be more coherent. Please kindly refer to our previous responses to locate the changes made.

Reviewer 2 Report

While this is a very interesting paper, there are some issues the authors should revise:

- the paper analyses the role of ecotourism to promote forest conservation in Bangladesh.

- while the introduction is well developed, and the research question is identified, the authors should engage with more examples of recent research.

- the first sentence of the paper describes the relevance of tourism industry, but data provided should be updated and also consider the pandemic situation.

- the authors should discuss why ecotourism “has often been criticized for being ineffective and/or harmful”.

- outdated data is provided in relation to the contribution of tourism in Bangladesh.

- the methodology is explained, however the authors should describe the methodological appropriateness of using a case study and provide examples of previous research to support the rationale of the method.

- the authors should expand the discussion with more recent research and the global relevance of the study.

- the authors should expand the opportunities for further research.

Author Response

REVIEWER 2

Comment: While this is a very interesting paper, there are some issues the authors should revise:

Response: We thank the reviewer for taking the time to thoroughly review our manuscript and offer their very useful feedback in helping us improve the quality of the manuscript.

Comment: the paper analyses the role of ecotourism to promote forest conservation in Bangladesh. while the introduction is well developed, and the research question is identified, the authors should engage with more examples of recent research.

Response: Thank you for your comment. As (newly) mentioned in the manuscript, there is a dearth of literature that assesses the impacts of ecotourism on forest coverage. This gap in knowledge has been outlined as follows:

“The application of GIS has also been recommended in the assessment of biodiversity and forest conservation practices [57–59]. Overall, however, there is a dearth of literature that empirically analyzes ecotourism impacts on forests. Brandt and Buckley [13] found only 17 studies, published between 2000 to 2018, which evaluated the potential of ecotourism for forest protection in biodiversity hotspots. The majority of the studies (14 of 17) evaluated forest change using satellite data, while the remaining used methods from social sciences (such as surveys and interviews). The archetypal case study for the negative forest outcomes of ecotourism, for instance, was illustrated by Liu et al. [60] in the analysis of the impacts of tourism on the Wolong Giant Panda Nature Reserve in southwest China. The study had revealed that there was an increase in deforestation after the Reserve had been implemented, and found that, surprisingly, deforestation had occurred more within the Reserve than outside. Similar studies were conducted during the aforementioned period in other ecotourism spots in China, and other developing nations like India, Nepal, Cambodia, Mexico, Belize, and Peru [13]. The most recent literature [61], focused on the land use and cover of the Shivpuri watershed in Nepal, also repeat a similar scene- a 110 ha reduction in forest coverage between 1999 to 2016. However, the application of GIS and/or RS technology to assess the effectiveness of biodiversity conservation strategies, such as in protected areas, have been less extensive, even more so in Bangladesh. Nevertheless, one study found that the Himchari National Park, a protected area has been degraded, fragmented, and converted severely into various land uses, where nearly half of the dense forest land was converted to other land uses in the period of 1977 to 2017 [62].” (1.4. Application of GIS in Ecotourism Research and Vegetation Cover Analysis, paragraph 3, page 4).

The latest data was sought as much as possible, and when not found, old data had to be used. Some more literature exists, as cited by Brandt and Buckley (2018), but they are too old, where most were conducted before 2015, and do not focus on forest coverage. In terms of old data in relation to the ecotourism sector in Bangladesh, it should be confessed that tourism, let alone ecotourism, is a field that is not very widely studied in Bangladesh (which has also contributed to the novelty of this research). Relevant literature was comprehensively scoped but recent studies on the topic could still not be found.

Nevertheless, the following literatures were newly added:

“....For instance, when a PES system was instituted at the Monarch Butterfly Biosphere Reserve in Mexico, more gains than losses were experienced in the closed cover density during 1999-2009 [16]... Several ecotourism initiatives, such as the Chitwan National Park in Nepal [13,19], are illustrations of ecotourism as a promising forest conservation tool. In the Chitwan National Park, satellite image analysis displayed regeneration of many forest patches after the introduction of a buffer zones program due to significant investment in plantation and forest-management initiatives.” (1.1. Ecotourism as a Tool for Sustainable Development and Biodiversity Conservation, paragraph 3, page 2).

“...The archetypal case study for the negative forest outcomes of ecotourism, for instance, was illustrated by Liu et al. [60] in the analysis of the impacts of tourism on the Wolong Giant Panda Nature Reserve in southwest China. The study had revealed that there was an increase in deforestation after the Reserve had been implemented, and found that, surprisingly, deforestation had occurred more within the Reserve than outside. Similar studies were conducted during the aforementioned period in other ecotourism spots in China, and other developing nations like India, Nepal, Cambodia, Mexico, Belize, and Peru [13]. The most recent literature [61], focused on the land use and cover of the Shivpuri watershed in Nepal, also repeat a similar scene- a 110 ha reduction in forest coverage between 1999 to 2016....” (1.4. Application of GIS in Ecotourism Research and Vegetation Cover Analysis, paragraph 3, page 4).

“...As discussed, deforestation rates have been observed to be higher within protected areas in other ecotourism spots around the world as well, such as the Wolong Giant Panda Nature Reserve [60]. The case study has also singled out disturbances by the local community within the Reserve as the primary cause of the destruction of the forest and panda habitat.” (4. Discussion and Recommendations, paragraph 3, page 16).

“...The importance of collaboration with the local community and other stakeholders has been recognized in other studies as well, such as in Gunung Leuser National Park of Indonesia by Hartoyo et al. [74], in order to retain its densely vegetated state. In a recent study [75] conducted in Chambok, Cambodia, community-based ecotourism was found to effectively lower deforestation, revealed through both satellite imagery analysis and surveys. Conservation can only be successful...” (4. Discussion and Recommendations, paragraph 4, page 16).

Comment: the first sentence of the paper describes the relevance of tourism industry, but data provided should be updated and also consider the pandemic situation.

Response: New data has been provided, and the sentence(s) has been re-written as follows: “The tourism industry has become one of the fastest-growing industries in the service sector, where between 2009 and (pre-pandemic) 2019, the real growth in international tourism receipts (54%) exceeded growth in world GDP (44%), contributing to USD 1,481 billion for total international tourism receipts alone [1]. Even though the COVID-19 pandemic had significantly lowered tourism across the world, having crossed a drop by 73% in international global tourist arrivals in 2020, international tourism experiences signs of a rebound in June and July 2021, attributed to the easing travel restrictions and the advancing global vaccination rollout [2].” (1.1. Ecotourism as a Tool for Sustainable Development and Biodiversity Conservation, paragraph 1, page 1).

Comment: the authors should discuss why ecotourism “has often been criticized for being ineffective and/or harmful”.

Response: The statement has been briefly discussed: “...but has often been criticized for being ineffective and/or harmful, initiating numerous environmental risks such as water pollution and old-growth deforestation due to the increasing reliance and usage of natural resources, especially forest products when the ecotourism spot is or nearby a forest [13–15]. In many case studies, activities related to tourism increased the demand for timber and fuelwood for the construction of new infrastructures, including housing [13]. The economic development of the area also acted as pull factors for migration and population growth, and many forested lands were cleared for other land uses [13].” (1.1. Ecotourism as a Tool for Sustainable Development and Biodiversity Conservation, paragraph 3, page 2).

Comment: outdated data is provided in relation to the contribution of tourism in Bangladesh.

Response: Tourism, let alone ecotourism, is a field that is not very widely studied in Bangladesh (which has also contributed to the novelty of this research). As responses from some of the previous comments read, there is a dearth of literature that empirically analyzes ecotourism impacts on forests. This issue is more marked when reviewing literature for ecotourism, vegetation coverage, and GIS mapping in the context of Bangladesh. The latest data was sought as much as possible, and where not found, old data had to be used. Relevant literature was comprehensively scoped but recent studies on the topic could still not be found.

Nevertheless, the following literatures were newly added:

 “...As discussed, deforestation rates have been observed to be higher within protected areas in other ecotourism spots around the world as well, such as the Wolong Giant Panda Nature Reserve [60]. The case study has also singled out disturbances by the local community within the Reserve as the primary cause of the destruction of the forest and panda habitat.” (4. Discussion and Recommendations, paragraph 3, page 16).

“...The importance of collaboration with the local community and other stakeholders has been recognized in other studies as well, such as in Gunung Leuser National Park of Indonesia by Hartoyo et al. [74], in order to retain its densely vegetated state. In a recent study [75] conducted in Chambok, Cambodia, community-based ecotourism was found to effectively lower deforestation, revealed through both satellite imagery analysis and surveys. Conservation can only be successful...” (4. Discussion and Recommendations, paragraph 4. page 16).

If there are specific pieces of data or literature that the reviewer would like us to find, remove, or replace, please do let us know and we will try our best to address them.

Comment: the methodology is explained, however the authors should describe the methodological appropriateness of using a case study and provide examples of previous research to support the rationale of the method.

Response: Thank you for this suggestion. A rationale for the method has been provided, as follows:

“          The application of GIS has also been recommended in the assessment of biodiversity and forest conservation practices [57–59]. Overall, however, there is a dearth of literature that empirically analyzes ecotourism impacts on forests. Brandt and Buckley [13] found only 17 studies, published between 2000 to 2018, which evaluated the potential of ecotourism for forest protection in biodiversity hotspots. The majority of the studies (14 of 17) evaluated forest change using satellite data, while the remaining used methods from social sciences (such as surveys and interviews). The archetypal case study for the negative forest outcomes of ecotourism, for instance, was illustrated by Liu et al. [60] in the analysis of the impacts of tourism on the Wolong Giant Panda Nature Reserve in southwest China. The study had revealed that there was an increase in deforestation after the Reserve had been implemented, and found that, surprisingly, deforestation had occurred more within the Reserve than outside. Similar studies were conducted during the aforementioned period in other ecotourism spots in China, and other developing nations like India, Nepal, Cambodia, Mexico, Belize, and Peru [13]. The most recent literature [61], focused on the land use and cover of the Shivpuri watershed in Nepal, also repeat a similar scene- a 110 ha reduction in forest coverage between 1999 to 2016. However, the application of GIS and/or RS technology to assess the effectiveness of biodiversity conservation strategies, such as in protected areas, have been less extensive, even more so in Bangladesh. Nevertheless, one study found that the Himchari National Park, a protected area has been degraded, fragmented, and converted severely into various land uses, where nearly half of the dense forest land was converted to other land uses in the period of 1977 to 2017 [62].” (1.4. Application of GIS in Ecotourism Research and Vegetation Cover Analysis, paragraph 3, page 4)

“... Most scholarly works have attempted to assess the flora and fauna biodiversity of the SBGE, but a spatiotemporal analysis of how the vegetation coverage has changed has not been attempted before for the SBGE. Accordingly, the objectives of this study were to explore how effectively ecotourism and consequently, a protected status, served as a forest conservation tool by assessing the vegetation cover changes from 1995 to 2020 in and around the SBGE by employing GIS-RS technology. Similar studies, even though few in number, produced over the years that assess the impacts of ecotourism on forest protection in biodiversity hotspots tended to measure the forest coverage change before and after tourism implementation [13], often complemented with some socio-economic data or social science methods and regression models to support the association. The Wolong Giant Panda case study, for example, represents a clear understanding of how coupling between human and natural systems vary across spatial units. The understanding of the complexities between human and natural systems is often hindered by the academic separation of ecological and social sciences [66]. Therefore, an interdisciplinary perspective comprising ecological and social sciences are crucial to study coupled human and natural systems [66]. However, as this study was conducted when the pandemic situation had been serious, with high rates of death and infectivity and intensified mobility restrictions (early 2020-early 2021), no primary data could be collected to complement the findings from the GIS-RS analysis (more on Section 3.3. Limitations). Thus, this study attempts to suggest and conceptualize plausible anthropogenic activities that may have contributed to the changes in vegetation by providing insights into the tourism practices in and around the SBGE, and the socio-economic activities of the local community via a comprehensive literature review of relevant studies surrounding the SBGE.” (1.5. Objectives, paragraph 1, pages 5).

Comment: the authors should expand the discussion with more recent research and the global relevance of the study.

Response: Please refer to our previous responses, where recent research was addressed. For the global relevance of the study, the following was added:

“In spite of its limitations, this study was able to effectively portray the inadequacies in current ecotourism initiatives in Bangladesh and echoed some of the major failures of ecotourism interventions in many of the developing nations. Consequently, similar policy implications, i.e. of community-based ecotourism and co-management practices, and sustainable land-use policies, should be translated for nations grappling with the same issue of declining forest coverage resulting from ecotourism interventions.” (4. Discussion and Recommendations, paragraph 10, page 18).

Comment: the authors should expand the opportunities for further research.

Response: Opportunities for future research has been expanded on, as follows:

“Even though unsustainable forms of urbanization and interventions from the local communities have been suggested as the major contributor behind the trend in vegetation coverage of the SBGE observed, the uncertainty causes a gap to still remain regarding the main cause behind the trend. Therefore, future research work needs to be directed towards understanding the primary socio-economic data and drivers of change. For instance, models incorporating elevation, climate data, and the income derived from tourism, and considerations of the changing political economy are some aspects that need further exploring. Moreover, remote sensing data with a higher spatial resolution are needed to detect smaller changes in the SBGE across shorter time intervals. More importantly, future ecotourism initiatives should integrate GIS-RS technologies in the decision-making processes, so that sensitive ecological areas can be properly delineated from urban settlements with adequate regions of buffer zones.” (4. Discussion and Recommendations, paragraph 8, page 17-18).

Round 2

Reviewer 1 Report

Dear Authors,

thanks for the revised version, the manuscript has been improved even if some suggestions from the reviewers were ignored.

Author Response

REVIEWER 1

Comment: Dear Authors,

thanks for the revised version, the manuscript has been improved even if some suggestions from the reviewers were ignored.

Response: Dear reviewer,

Thank you for taking the time to meticulously review our manuscript and offering such useful feedback in helping us improve the quality of the manuscript.

We apologize if it seemed like some of the suggestions were ignored; we did try our best to address each and every comment or provide reasoning behind those that we could not incorporate. Regardless, we are very glad that you feel that the manuscript seems to have improved. This wouldn’t have been possible without your time and effort. We thank you again.

Reviewer 2 Report

While the authors have revised the paper, the methodological section is not improved. The authors should move the methodological rationale to section 2, and they should also add brief paragraphs to describe the sections between 1 and 1.1, 2 and 2.1, 3 and 3.1.

Author Response

REVIEWER 2

Comment: While the authors have revised the paper, the methodological section is not improved. The authors should move the methodological rationale to section 2, and they should also add brief paragraphs to describe the sections between 1 and 1.1, 2 and 2.1, 3 and 3.1.

Response: Thank you for the further clarification on the improvements required for the methodology section.

We shifted some of the paragraphs from the Introduction, as suggested, and provided the rationale for the methodology in the methods section.

Section 2 now reads as follows (2. Methodology, page 5-6, paragraph 1-2):

“Overall, there is a dearth of literature that empirically analyzes ecotourism impacts on forests. Brandt and Buckley [20] found only 17 studies, published between 2000 to 2018, which evaluated the potential of ecotourism for forest protection in biodiversity hotspots. The majority of the studies (14 of 17) evaluated forest change using satellite data, while the remaining used methods from social sciences (such as surveys and interviews) and regression models to support the association. The Wolong Giant Panda case study, for example, represents a clear understanding of how the coupling between human and natural systems varies across spatial units. The understanding of the complexities between human and natural systems is often hindered by the academic separation of ecological and social sciences [65]. Therefore, an interdisciplinary perspective comprising ecological and social sciences is crucial to study coupled human and natural systems [65]. As mentioned, this study employs GIS-RS techniques to assess the forest conservation initiatives in the SBGE. However, as this study was conducted when the pandemic situation had been serious, with high rates of death and infectivity and intensified mobility restrictions (early 2020-early 2021), no primary data could be collected to complement the findings from the GIS-RS analysis (more on Section 3.3. Limitations).

This section describes the methods that were undertaken by this study in detail. Figure 1 below provides the summary of the workflow of the vegetation coverage analysis conducted for this study.

Section 2.1....”

Moreover, as suggested, some brief introductory paragraphs were provided for the sections.

For the introduction, in order to provide a brief overview of the generalized topic, we have moved the paragraphs on tourism in general under introduction and before Section 1.1, and a small sentence as an introductory to Bangladesh.

The introduction now reads as follows (1. Introduction, page 1-2, paragraphs 1-3):

“The tourism industry has become one of the fastest-growing industries in the service sector, where between 2009 and (pre-pandemic) 2019, the real growth in international tourism receipts (54%) exceeded growth in world GDP (44%), contributing to USD 1,481 billion for total international tourism receipts alone [1]. Even though the COVID-19 pandemic had significantly lowered tourism across the world, having crossed a drop by 73% in international global tourist arrivals in 2020, international tourism experiences signs of a rebound in June and July 2021, attributed to the easing travel restrictions and the advancing global vaccination rollout [2].

Tourism can be of different kinds, usually deriving its characteristics from the intentions of the tourist. One such facet of tourism is ecotourism. The term ‘ecotourism’ was first coined by Ceballos-Lascuráin in the early 1980s, where he defined it as ‘traveling to relatively undisturbed or uncontaminated natural areas with the specific objective of studying, admiring, and enjoying the scenery and its wild plants and animals, as well as any existing cultural manifestations (both past and present) found in these areas’ [3] (p. 17). Ecotourism primarily emerged as the need for sustainable tourism was recognized since mass tourism often ensued uneven development and high social and environmental costs [4]. It is increasingly being viewed as a potential tool that can bring about sustainable development [5–7], as it is concerned with sustainable forms of tourism that take place in the natural areas, and promotes environmental conservation, environmental awareness, travelers’ responsibility, and active community participation [8]. It allows for the reconciliation of both economic growth and environmental wellbeing, as ecotourism generates revenues while simultaneously encouraging initiatives for the conservation and the management of biodiversity [9,10]. Therefore, ecotourism also has wide-ranging implications in global biodiversity conservation initiatives.

Ecotourism can serve as an important tool for sustainable development, especially in developing nations that possess impressive biodiversity hotspots [4,11–13]. For instance, Bangladesh, a South Asian developing country with a population of over 160 million people, derives its potential for ecotourism by possessing several world-famous natural sites, such as the Sundarban and the Cox’s Bazar, where it boasts not only are-as containing spectacular jungles rich in wildlife, waterfalls, rivers, and hilly land-scapes but also several cultural heritage sites [14,15]. Despite such positive connotations, ecotourism still remains a highly contentious concept [4], particularly due to its reliance on market-based conservation making [16,17]. As the implications of such limitations are far more significant in nations that are struggling to operationalize sustainable development, it is important to assess the impacts of ecotourism interventions in such countries. This section starts by providing a brief overview of how eco-tourism can be utilized to attain biodiversity conservation, revealing both its strengths and weaknesses in the endeavor. It then introduces readers to the study area of interest and delves into discussions of how remote sensing and spatial analytical techniques have been contributing to ecotourism research, especially in developing nations. Finally, it outlines the objectives of this study and provides insights into the novelty of this research.

  • Ecotourism as a Tool for Biodiversity Conservation...”

Because sustainable development has been addressed in the introduction, we have removed it from the title of section 1.1, which now reads “1.1. Ecotourism as a Tool for Biodiversity Conservation”.

Moreover, because of the shifts made in the paragraphs, changes can be perceived in: 1.2. The Untapped Potential of Ecotourism in Bangladesh (page 3, first paragraph, first line), 1.4. Application of GIS in Ecotourism Research and Vegetation Cover Analysis (page 4, between paragraphs 2 and 3), and 1.5. Objectives (page 5, in the middle of first paragraph).

Section 3 begins with the sentence below now, followed by its subsections (Section 3. Results, page 9, paragraph 1):

“The results obtained from the temporal-spatial analysis of the vegetation coverage of the SBGE and their corresponding accuracy assessment are presented in the following subsections. The section concludes with a brief discussion on the limitations of the study.

3.1. Vegetation Cover Change of SBGE from 1995 to 2020...”

We thank the reviewer again for assisting us with their valuable feedback for the second round of revision. We hope we were now able to successfully address all of their comments.